# Knowledge and Practices of Digestive Surgeons concerning Specialized Nutritional Support in Cancer Patients: A Survey Study

**DOI:** 10.3390/nu14224764

**Published:** 2022-11-11

**Authors:** Manuel Durán-Poveda, Alejandro Suárez-de-la-Rica, Emilia Cancer Minchot, Julia Ocón-Bretón, Andrés Sánchez-Pernaute, Gil Rodríguez-Caravaca

**Affiliations:** 1Department of Digestive and General Surgery, Hospital Universitario Rey Juan Carlos, 28933 Móstoles, Spain; 2Department of Medical Specialties and Public Health, Universidad Rey Juan Carlos, 28922 Alcorcón, Spain; 3Department of Anesthesiology and Surgical Critical Care, Hospital Universitario, La Paz, 28046 Madrid, Spain; 4Service of Endocrinology and Nutrition, Hospital Universitario de Fuenlabrada, 28942 Fuenlabrada, Spain; 5Service of Endocrinology and Nutrition, Hospital Clínico Universitario “Lozano Blesa”, 50009 Zaragoza, Spain; 6Department of Digestive and General Surgery, Hospital Clínico San Carlos, 28040 Madrid, Spain; 7Service of Preventive Medicine, Hospital Universitario Fundación Alcorcón, 28922 Alcorcón, Spain

**Keywords:** nutrition, cancer patients, digestive surgeons, knowledge, survey

## Abstract

A survey study based on a 21-item questionnaire was conducted to assess knowledge and practices of digestive surgeons focused on nutritional support in gastrointestinal cancer patients. At least 5 staff digestive surgeons from 25 tertiary care hospitals throughout Spain were invited to participate and 116 accepted. Malnutrition was correctly defined by 81.9% of participants. In patients undergoing major abdominal surgery, 55.2% considered that preoperative nutritional support is indicated in all patients with malnutrition for a period of 7–14 days. For the diagnosis of malnutrition, only 18.1% of participants selected unintentional weight loss together with a fasting or semi-fasting period of more than one week. Regarding the advantages of enteral infusion, 93.7% of participants considered preservation of the integrity of the intestinal mucosa and barrier function, and in relation to peripheral parenteral nutrition, 86.2% selected the definition of nutrient infusion through a peripheral vein and 81.9% its indication for less than 7 days. Digestive surgeons had a limited knowledge of basic aspects of clinical nutrition in cancer patients, but there was some variability regarding clinical practice in individual cases. These findings indicate the need to develop standardized clinical protocols as well as a national consensus on nutrition support in cancer patients.

## 1. Introduction

Malnutrition is a common problem in patients with cancer, which may be present in up to 85% of patients with certain types of tumors (e.g., pancreas) [1]. Malnutrition can negatively affect the outcome of treatment and is an important causative factor of impaired quality of life. The proportion of cancer patients with weight loss at diagnosis ranges between 15% and 40% [2], although the incidence of malnutrition increases as malignancy progresses, with severe weight loss in about 80% of patients with advanced disease, and depletion of skeletal muscles as a key feature of cancer-associated cachexia [3]. Malnutrition and cachexia are multifactorial and lead to functional impairment, increased chemotherapy toxicity, impaired immune responses, complications from cancer surgery, and increased morbidity and mortality [4,5,6]. Malnutrition is also associated with increases in healthcare-related costs [7]. Beyond the acute phase of treatment, poor nutrition negatively influences quality of life and well-being in cancer survivors [8].

The evidence-based guidelines of the European Society for Clinical Nutrition and Metabolism (ESPEN) for nutritional care in cancer patients [9] emphasize three essential steps: (1) screening all patients with cancer for nutritional risk early in the course of the disease, (2) expanding nutrition-related assessment practices with measures of body composition, inflammatory biomarkers, resting energy expenditure, and physical function, and (3) using multimodal nutritional interventions with individualized plans focused on increasing nutritional intake, lessening inflammation and hypermetabolic stress, and increasing physical activity. However, current evidence-based recommendations including the American Society of Parenteral and Enteral Nutrition (ASPEN) [10] and guidelines for nutrition and weight management in cancer patients [9,11,12,13] are not routinely translated into clinical practice. Although the importance of nutrition in surgical oncology is well established, malnutrition is frequently overlooked, screening techniques are not always standardized, and there is an important variability in the management of nutrition in daily practice. On the other hand, evidence-based guidelines may not provide answers for some controversial aspects and clinical scenarios in terms of decision making that healthcare professionals must deal with in the care of their individual patients [14].

Survey studies among specialists focused on nutritional support have shown that nutrition in general is largely a neglected issue, and that awareness and knowledge of nutrition therapy need to be improved in order to optimize nutrition care in the management of cancer patients [15,16,17,18,19]. However, none of these previous studies have been carried out in the framework of digestive surgeons treating patients undergoing surgery for gastrointestinal cancer. Therefore, the objective of the present survey study was to assess knowledge and practices of digestive surgeons concerning specialized nutritional support in gastrointestinal cancer patients.

## 2. Materials and Methods

### 2.1. Design, Participants, and Procedures

This survey study was conducted in parallel to a prospective nationwide multicenter and exploratory study (PREMAS study, Prevalence of Malnutrition in gastrointestinal Surgical oncology patients), the primary objective of which was to determine the prevalence of malnutrition in adult patients undergoing elective abdominal surgical procedures for the treatment of malignant tumors of the gastrointestinal tract. Staff surgeons from 23 tertiary care hospitals throughout Spain participated in the PREMAS study. Hospitals were selected at random by conglomerate sampling, in which each conglomerate was one the Spanish autonomous communities.

For the purpose of the present survey, a multidisciplinary expert panel (scientific committee) composed of two digestive surgeons, one anesthetist, one specialist in preventive medicine, and two specialists in endocrinology and nutrition was established. These specialists were authors of relevant research publications and were renowned professionals in the care of oncological patients, with expertise in digestive neoplasms and nutrition. The members of the scientific committee participated in the development of the questionnaire and supervised the progression of the study, including recruitment of participants and results of data analysis.

The final questionnaire was composed of 21 items and divided into two main sections. The first section included 13 items with general questions regarding the surgeons’ knowledge of nutrition (definition of malnutrition, nutritional assessment, nutritional support, and access routes). The second section included 8 items with questions focused on daily practice in individual case scenarios. The study questionnaire is described in the Appendix A. At least 5 surgeons from each of the 23 participating hospitals in the PREMAS study were invited to complete the questionnaire. Participation in the study was anonymous, voluntary, and unpaid. Participants who accepted to take part in the study were provided a printed copy of the questionnaire, which was returned after completion.

Since this study was based on a survey answered by surgeons specialized in gastrointestinal cancer, no approval was obtained from the ethics committees of the participating centers; however, the study was conducted according to principles of the Helsinki declaration.

### 2.2. Statistical Analysis

Sample size was estimated with a 95% confidence level, accuracy of 10%, expected compliance of 50%, and envisaged losses to follow-up of 10%. A total of 107 participants were thus deemed necessary. Descriptive statistics included frequencies and percentages for categorical variables. Data were analyzed using the SAS statistical program (Statistical Analysis Systems, SAS Institute, Cary, NC, USA) version 9.1.3 for Windows.

## 3. Results

Of a total of 125 potential participants (5 surgeons from each of the 25 participating hospitals in the PREMAS study), 116 (92.8%) agreed to take part in the study and completed the questionnaire.

### 3.1. Surgeons’ General Knowledge of Nutrition

Responses to the first part of the questionnaire are shown in Table 1.

A large percentage of participants (81.9%) selected the definition of malnutrition as “a nutritional state due to a decrease in the intake or incorporation of nutrients that leads to an alteration in body composition resulting in a reduction of functional and mental capacity, as well as worsening of clinical outcome”. A decrease in serum albumin and prealbumin were the parameters that most accurately diagnosed malnutrition according to 74.1% of participants, whereas a body mass index (BMI) < 20 kg/m^2^ was selected by 8.6%. However, the correct answer of unintentional weight loss together with a fasting or semi-fasting period of more than one week was selected by only 12.1% of participants. Malnutrition that occurs in surgical patients was attributed to protein malnutrition and related to the acute disease by 34.5% and 31% of participants, respectively, although only 18.1% considered malnutrition related to chronic disease with inflammation. About 50% of participants refused using the Charlson comorbidity index as a tool for nutritional screening. In addition, 60.3% of participants selected “a form of feeding in which the physiological route of food access to the digestive system is modified by artificial means, or the physiochemical composition of the nutritional mixture is modified” as the most appropriate definition of enteral nutrition. In the case of total parenteral nutrition (TPN), the tip of the central venous catheter should be placed in the superior cavoatrial junction according to 69% of participants and in the subclavian vein according to 21.6%.

In patients undergoing major abdominal surgery, 55.2% of participants considered that preoperative nutritional support is indicated in all patients with malnutrition for a period of 7–14 days, and 37.9% that preoperative nutritional support for 7 days improves the patient’s nutritional status. In addition, 62.1% considered that TPN should be maintained for at least 5–7 days to obtain any postoperative clinical benefit. Almost all participants (95.7%) considered that oral diet should be started in the first 24 h after major abdominal surgical procedures. Immunonutrition was recommended to reduce postoperative complications and length of hospital stay by 69.8% of participants and in malnourished patients with cancer by 67.2%. The indications of postoperative nutritional support are shown in Figure 1. A total of 40.5% of participants indicated TPN in the case of complications associated with intestinal failure when nutritional requirements are not fulfilled by oral intake in 7–10 days in normally nourished patients.

Regarding the advantages of enteral infusion of nutrients, 93.7% of participants considered preservation of the integrity of the intestinal mucosa and the digestive barrier function, and in relation to peripheral parenteral nutrition, 86.2% selected the definition of nutrient infusion through a peripheral vein and 81.9% its indication for less than 7 days.

### 3.2. Daily Practice and Case-Based Scenarios

As shown in Table 2, there was more variability in the percentages of response in this section of the questionnaire. More than half of participants (58.6%) would indicate preoperative TPN for 7–10 days only in a patient with severe malnutrition and intestinal occlusion requiring surgery. In a patient with esophageal cancer without weight loss undergoing esophagectomy, preoperative nutritional supplements were recommended by 48.3% of participants and enteral nutrition through jejunostomy by 23.3%. In addition, 59.5% of participants would refer a patient with pancreatic cancer and weight loss to a service of endocrinology and nutrition, and 63.8% would perform a jejunostomy for enteral nutrition during neoadjuvant therapy in a patient with locally advanced and obstructive gastric cancer. After laparoscopic right hemicolectomy and a paralytic ileus for 5 days, 45.7% of participants believed that ileus will be resolved and peripheral parenteral nutrition may be sufficient, but 20.7% would indicate TPN for at least 5–7 days. On the other hand, nutritional screening would be performed by 79.3% of participants in an obese patient with colon cancer.

In relation to measures of the Enhanced Recovery After Surgery (ERAS) protocol and preoperative recommendations in a patient with sigmoid rectal cancer, there were different percentages of responses, although “No mechanical bowel preparation, 8-h fasting for solids and 2-h for liquids and starting oral fluid intake at 6 h after surgery” accounted for the highest percentage (32.8%) (Figure 2). Finally, 83.6% of participants considered that all three measures of the ERAS protocol (laparoscopic surgery as the most effective method to reduce pain and postoperative ileus, multimodal epidural analgesia, and infiltration of laparoscopic ports or bilateral transversus abdominis plane block to reduce the need of rescue opioids) could have changed the postoperative clinical course of a patient with nausea, vomiting, prolonged ileus, and delay of oral feeding in whom a morphine pump was indicated because of pain after laparoscopic right hemicolectomy.

## 4. Discussion

The integration of nutrition into the overall management of cancer patients, particularly in those with malignant tumors of the digestive tract, is one of the key aspects of perioperative care. This survey study was conducted to evaluate knowledge and clinical practice of nutrition in surgical departments in Spain. From the potential total number of addressed physicians, 5 for each of the 23 centers participating in the PREMAS study, 116 answered the questionnaire, which indicates the interest among specialists regarding the benefits of appropriate nutritional support in oncological patients.

The surgeons’ knowledge of different questions on definition of malnutrition, nutritional screening, enteral and TPN, and preoperative nutritional support was generally satisfactory. A high percentage of participants (81.9%) correctly defined malnutrition and enteral nutrition. On the other hand, some aspects of knowledge and awareness of nutrition in cancer patients could be improved. It should be noted that only 18.1% of participants correctly classified malnutrition in surgical patients due to unintentional weight loss together with a fasting or semi-fasting period of more than one week. In relation to nutritional screening, 26.7% did not identify which questionnaires should be used, although 54.3% identified that the Charlson comorbidity index was not appropriate. However, specific questionnaires for nutritional screening such as the Malnutrition Screening Tool (MST), Malnutrition Universal Screening Tool (MUST), and the Mini-Nutritional Assessment Short-Form (MNA-SF) would not have been used by 3.4%, 5.2%, and 10.3% of participants, respectively.

In relation to immunonutrition, 25% of participants considered that evidence was insufficient to recommend its use in the preoperative period only. It has been suggested that lack of awareness regarding positive data for immunonutrition impedes usage [15]. In a meta-analysis of 61 randomized controlled trials (RCT) to determine the efficacy of immunonutrition versus standard nutrition in cancer patients treated with surgery, perioperative immunonutrition reduced the rate of wound infection and in malnourished patients shortened the hospitalization time but did not reduce all-cause mortality [20]. In another systematic review and meta-analysis of nine RCTs involving 966 patients undergoing hepatectomy, immunonutrition significantly reduced the incidences of overall postoperative complications and shortened the length of hospitalization [21].

Other survey studies have also evaluated knowledge and practices of different specialists concerning nutrition therapy in cancer patients and in surgical patients, but results are difficult to compare due to differences in the selection and characteristics of the items included in the ad hoc questionnaires. In a global survey designed by the European Society of Surgical Oncology (ESSO) and its young alumni club (EYSAC) to audit surgeons’ practice of nutritional assessment in different surgical oncology specialties, preoperative information regarding unplanned weight loss was “always” collected by only 16.3% to 44.4% of a total of 377 participants [19]. In a study of 109 Turkish medical oncologists, 43.1% received clinical nutrition education and 40% followed oncology sections in the ESPEN guidelines, and in both cases, having nutrition education and following ESPEN guidelines correlated to higher knowledge score calculated according to case scenario questions [22]. In UK specialist oncological trainees, lack of guidelines, knowledge, and time were identified as barriers for the ability to identify factors that place cancer patients at risk for malnutrition [23]. Poor implementation of evidence-based nutrition practices was also reported in a nationwide nutritional practice survey of U.S. gastrointestinal and oncologic surgeons [15]. In our study, nutrition education or adherence to evidence-based guidelines was not analyzed.

A clinically relevant finding of the present survey is the scarce knowledge of nutritional screening; therefore, the implementation of measures addressed to include routine nutritional screening in patients undergoing major surgery appears to be an area of improvement. In a survey conducted in 173 Swiss and Austrian surgical departments, only 20% reported routine nutritional screening, with financial and logistic restrictions as the main reasons of non-compliance [16].

In the second part of the questionnaire in which questions were focused on examples of individual cases, the percentages of responses were more varied. Preoperative TPN in a patient with intestinal occlusion requiring surgery was indicated only in the presence of malnutrition and for at least 7–10 days by 58.6% of participants, whereas 17.2% would indicate TPN always independently of whether surgery will be performed in the next 48 h. However, the precise role of TPN in the management of oncological patients with intestinal occlusion is not well defined. TPN in patients with advanced cancer is safe, and in a study of 55 patients with intestinal occlusion and peritoneal carcinomatosis, those who showed a response to chemotherapy and received home TPN together with active antitumoral treatment showed increased survival [24].

In the case of pancreatic cancer and marked weight loss, almost 60% of participants would refer the patient to the endocrinology/nutrition service for appropriate treatment, but 16.4% would schedule oral supplements only during the tumor staging process. In a survey of surgeons aimed at assessing their knowledge on nutritional support in patients undergoing cytoreductive surgery and hyperthermic intraperitoneal chemotherapy, although there was access to a dietitian on an inpatient and outpatient basis, only 32.7% reported to always consult a dietitian [18]. In a systematic review and meta-analysis of five RCTs with 737 cancer patients who were malnourished or at risk of malnutrition, dietary advice given by dietitians compared to usual care was associated with significant improvement in energy intake and quality of life [25].

Regarding the attitude in the presence of a prolonged paralytic ileus after laparoscopic right hemicolectomy, 45.7% of participants selected conservative management and 20.7% would indicate TPN for 5–7 days. Prolonged postoperative ileus occurs in up to 25% of patients undergoing major elective abdominal surgery and is associated with a higher risk of developing postoperative complications and prolongs hospital stay, increasing direct healthcare costs. Conservative management with nasogastric decompression, adequate fluid replacement therapy, regular ambulation, and parenteral nutrition if the patient is unable to tolerate oral intake for more than 7 days postoperatively has been recommended, although evidence-based guidelines still remain unclear [26].

In an obese patient diagnosed with colon cancer, 79.3% of participants would perform nutritional screening as in all patients. Despite the fact that malnutrition predicts poorer clinical outcomes in patients with cancer, nutritional screening at the time of diagnosis is not routinely performed. In a two-round Delphi study involving 52 specialists with experience in nutritional support in cancer patients conducted in Spain, 62.9% of participants considering the screening to assess the risk of malnutrition was performed in less than 30% of patients [14]. On the other hand, although a number of malnutrition screening tools exist, the best screening tool for cancer patients remains to be established. In a systematic review of 42 studies describing 15 markers of malnutrition, no tool was identified as appropriate to screen for malnutrition, as distinct from inflammatory causes of weight loss [27].

Finally, albumin and prealbumin are parameters longer used in surgery to identify changes in nutritional status of the patients. However, they are not accurate enough to define malnutrition, but in the present study, a decrease in serum albumin and prealbumin were the parameters that most accurately diagnosed malnutrition according to 74.1% of participants. The guidelines of the European Society for Clinical Nutrition and Metabolism (ESPEN) [28] indicate that these parameters are more related to prognosis for postoperative complications and impaired nutritional status, although a persistently low or even decreasing or increasing serum albumin concentration may be indicative of whether recovery is successful or not. In the last updated ESPEN clinical guidelines, the working group agreed that hypoalbuminemia reflects a disease-associated catabolism and disease severity rather than undernutrition [29].

The present survey had a high response rate, but the lack of acquiring a randomized sample group is a limitation to generalize data to all Spanish surgeons practicing elective gastrointestinal procedures in cancer patients. The presentation of descriptive statistics only is also a limitation of the study, and it would have been interesting to perform further analysis to assess differences in responses according to characteristics of participants (such as age, sex, years of practice, participation in previous training activities or research projects, surgeon’s subspecialty, working with a dietitian, etc.). However, the sample size could be still large enough to have a sound opinion of the present situation. Strengths of the study include the homogeneity of participants and the fact that hospitals from the entire national territory were represented. It should also be considered that the surgeons who completed the questionnaire may have had a higher average degree of interest in nutritional aspects related to the management of oncological patients, which may be associated with the need for a more accurate and proper level of awareness and knowledge in nutrition than other digestive surgeons in general.

## 5. Conclusions

The present results provide evidence of a general limited knowledge of basic aspects of clinical nutrition in cancer patients among gastrointestinal surgeons who participated in the study. Due to the possibility of a selection bias, with participants being those more interested in the topic of the survey, limitations of knowledge may be under-reported. Areas needing improvement included adequate knowledge of nutritional screening, causes of malnutrition in the oncological gastrointestinal patient, indications of postoperative nutritional support, and use of postoperative TPN. However, some variability was detected in the classification of malnutrition that occurs in the surgical patient, indication of preoperative nutritional support, or recommendations of ERAS programs. An important variability was also found regarding clinical practice in individual case scenarios. These findings indicate the need to develop standardized clinical protocols as well as a national consensus on nutrition support in cancer patients.

## Figures and Tables

**Figure 1 nutrients-14-04764-f001:**
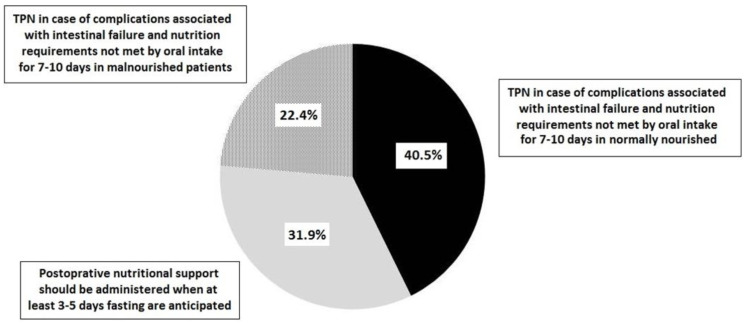
Percentages of responses to the question “which of the following statements regarding postoperative nutritional support is correct?” (TPN: total parenteral nutrition) (no answer 5.2%).

**Figure 2 nutrients-14-04764-f002:**
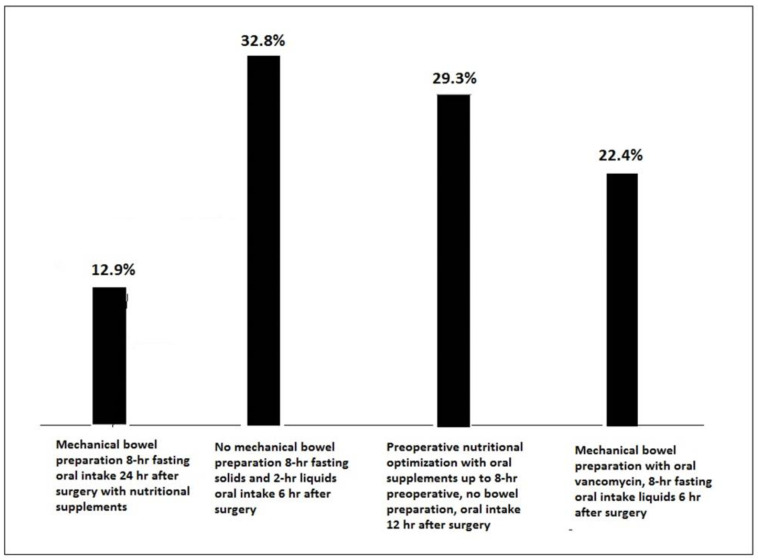
Preoperative recommendations in the use of an ERAS protocol for a patient with sigmoid rectal cancer (hr: hour).

**Table 1 nutrients-14-04764-t001:** Surgeons’ general knowledge of nutrition.

Items of the Questionnaire	Number of Responses (%)
**What do you consider that is the most appropriate definition of malnutrition?**	
− Is the nutritional state due to a decrease in the intake or incorporation of nutrients that leads to an alteration n body composition resulting in a reduction of functional and mental capacity, as well as worsening of clinical outcome	95 (81.9)
− Is the one that affects in a very special way to a particular group, such as hospitalized subjects in which disability and disease are common, taking an own entity	1 (0.9)
− Is that in which there is a decrease of albumin and total proteins due to any intercurrent process	10 (8.6)
− None of them	9 (7.8)
Don’t know/no answer	1 (0.9)
**In your opinion, with what parameters is malnutrition most accurately diagnosed?**	
− With the decrease of albumin and prealbumin	86 (74.1)
− With unintentional weight loss together with a fasting or semi-fasting period of more than one week	14 (12.1)
− With a body mass index < 20 kg/m^2^	10 (8.6)
− With a decrease of plasma levels of cholesterol and lymphocytes	4 (3.4)
Don’t know/no answer	2 (1.7)
**In your opinion, how malnutrition that occurs in the surgical patient should be classified?**	
− Malnutrition related to the chronic disease with inflammation	21 (18.1)
− Malnutrition related to fasting	10 (8.6)
− Malnutrition related to the acute disease	36 (31.0)
− Protein malnutrition	40 (34.5)
Don’t know/no answer	9 (7.8)
**Which of the following questionnaires would you NOT use for nutritional screening?**	
− MST (Malnutrition Screening Tool)	4 (3.4)
− MUST (Malnutrition Universal Screening Tool)	6 (5.2)
− Charlson	63 (54.3)
− MNA-SF (Mini-Nutritional Assessment Short-Form)	12 (10.3)
Don’t know/no answer	31 (26.7)
**What definition of enteral nutrition do you consider most appropriate?**	
− Form of feeding in which the physiological route of the access of food to the digestive system is modified by artificial means	9 (7.8)
− Form of feeding in which the physiological route of the access of food to the digestive system is modified by artificial means, or the physicochemical composition of the nutritional mixture	70 (60.3)
− Form of feeding in which the physicochemical composition of the nutritional mixture is modified	22 (19.0)
− None of them	15 (12.9)
**In your opinion, where should the tip of a central venous catheter be placed in order to start total parenteral nutrition?**	
− In the subclavian vein	25 (21.6)
− In the cephalic vein	1 (0.9)
− In the superior cavoatrial junction	80 (69.0)
− In the brachiocephalic trunk	9 (7.8)
Don’t know/no answer	1 (0.9)
**In relation to preoperative nutritional support in patients undergoing a major abdominal surgical procedure, which of the following statements do you consider to be true?**	
− It is indicated in all patients with severe malnutrition for a period of 7–14 days, even if oncological surgery has to be delayed	64 (55.2)
− Surgery should never be delayed to improve the nutritional status if the patient has an oncological disease	1 (0.9)
− It is only indicated in patients who cannot be nourished by the oral route	1 (0.9)
− 7 days of preoperative nutritional support improves the nutritional status (body composition and albumin levels)	44 (37.9)
Don’t know/no answer	6 (5.2)
**How long do you consider it is necessary to maintain total parenteral nutrition (TPN) to achieve any postoperative clinical benefit?**	
− At least 3 days	14 (12.1)
− At least 5–7 days	72 (62.1)
− More than 7 days	22 (19.0)
− At least 48 h	5 (4.3)
Don’t know/no answer	3 (2.6)
**What do you consider that are the advantages of enteral infusion of nutrients in the early postoperative period? Mark all options that apply**	
− To preserve the integrity of the intestinal mucosa (structure and function) and, therefore, the gastrointestinal barrier	109 (93.7)
− To improve blood flow and mesenteric oxygenation	68 (58.6)
− To reduce bacterial translocation and maintain immunocompetence	93 (80.2)
**In your opinion, which of the following statements regarding postoperative nutritional support is correct?**	
− TPN should be routinely administered postoperatively in patients with major gastrointestinal surgery	0
− Nutritional support after surgery should be administered in those patients in whom fasting times of at least 3–5 days are anticipated	37 (31.9)
− TPN is indicated postoperative when complications occur in association with intestinal failure, which make it foreseeable that the patient will be unable to meet their nutritional requirements through oral (enteral) intake for a period of 7–10 days in case of being malnourished	26 (22.4)
− TPN is indicated postoperative when complications occur in association with intestinal failure, which make it foreseeable that the patient will be unable to meet their nutritional requirements through oral (enteral) intake for a period of 7–10 days if he/she is normally nourished	47 (40.5)
Don’t know/no answer	6 (5.2)
**In most patients, when do you consider that oral diet should be started after major abdominal surgery?**	
− In the first 24 h	111 (95.7)
− When there is passage of feces	0
− When there is no leak in the intestinal transit	2 (1.7)
− After 5 days because it is considered to be the most appropriate period for the protection of sutures	1 (0.9)
Don’t know/no answer	2 (1.7)
**Regarding immunonutrition, add those statements that in your opinion are correct. Mark all options that apply**	
− Currently there is no clear evidence to recommend its use only in the preoperative period	29 (25.0)
− Its administration is recommended in the peri and postoperative period of malnourished patients with cancer	78 (67.2)
− Its use is associated with a significant reduction of postoperative complications and length of hospital stay	81 (69.8)
− It is not cost-effective	2 (1.7)
Don’t know/no answer	6 (5.2)
**In relation to peripheral parenteral nutrition, select those statements that in your opinion are correct. Mark all options that apply**	
− Nutrients are infused into the bloodstream through a peripheral vein	100 (86.2)
− With this type of nutritional support, 100% of the patient’s nutritional requirements are not usually covered	87 (75.0)
− It is indicated for periods of less than 7 days	95 (81.9)
− It is indicated in those patients who require fluid restriction, e.g., heart or liver failure	10 (8.6)

**Table 2 nutrients-14-04764-t002:** Surgeons’ opinions regarding clinical practice and individual case scenarios.

Items of the Questionnaire	Number of Responses (%)
**In a patient presenting with intestinal occlusion requiring a surgical operation, when would you indicate preoperative TPN?**	
− Always, regardless of whether he/she is going to be operated in the next 48 h	20 (17.2)
− Only in patients with mild malnutrition and for a period of at least 5 days	12 (10.3)
− Only in patients with severe malnutrition and for a period of at least 7–10 days	68 (58.6)
− In all patients regardless their nutritional status for at least 7 days	9 (7.8)
Don’t know/no answer	7 (6.0)
**In a patient with carcinoma of the esophagus without preoperative weight loss who will undergo esophagectomy, which of the following attitudes do you consider that are correct?**	
− Is a patient without preoperative malnutrition, so that postoperative nutritional support is not required	0
− Because it is expected to maintain the patient in fasting conditions for at least 5–7 postoperative days, it is recommended to administer preoperative nutritional supplements	56 (48.3)
− Due to a presumed prolonged lack of postoperative oral intake, it is recommended to administer TPN for at least 7–10 days	21 (18.1)
− In any case, enteral nutrition through jejunostomy would be indicated	27 (23.3)
Don’t know/no answer	12 (10.3)
**According to the protocol of your hospital, what is the procedure to follow when faced with a patient with cancer of the pancreas and weight loss of 20 kg in the last 3 months and BMI of 18 kg/m^2^?**	
− Schedule oral supplements from the first visit to the clinic, while the staging study is completed	19 (16.4)
− Make a nutritional screening by the own surgeon and to decide whether administration of oral supplements is indicated	11 (9.5)
− Refer to the Endocrinology/Nutrition consultation to assess the nutritional status and decide on the appropriate treatment	69 (59.5)
− Admit the patient to the hospital for TPN while tumor stating evaluation is being completed	8 (6.9)
Don’t know/no answer	9 (7.8)
**When faced with a patient with obstructive and locally advanced gastric cancer, indicate the attitude that seems most appropriate to you**	
− An endoscopic endoprosthesis should be placed to allow correct oral feeding during neoadjuvant treatment	26 (22.4)
− Curative surgery should be consider as the first step and subsequent adjuvant treatment	10 (8.6)
− During staging laparoscopy, jejunostomy should be performed in order to administer enteral nutrition while neoadjuvant treatment is used	74 (63.8)
− The patient is candidate for home TPN while receiving neoadjuvant treatment	2 (1.7)
Don’t know/no answer	4 (3.4)
**A patient who had undergone laparoscopic right hemicolectomy presents a paralytic ileus during 5 days, what would be the attitude to follow?**	
− It is expected that it would be resolved in a short time, so that peripheral parenteral nutrition may be sufficient	53 (45.7)
− TPN for at least 5–7 days	24 (20.7)
− Once occlusion by a mechanical cause has been excluded, the introduction of oral feeding is the best method for preventing paralytic ileus	17 (14.6)
− After a right hemicolectomy, paralytic ileus is frequent, so that preoperative supplementation is indicated even in normonourished patients	19 (16.4)
Don’t know/no answer	3 (2.6)
**When faced with an obese patient diagnosed with colon cancer, what would be your attitude in the consultation?**	
− I would indicate a diet because obesity increases the risk of postoperative complications	3 (2.6)
− I would not pay attention to the nutritional status, but I would pay attention to possible anemia, which must be optimized prior to surgery	1 (0.9)
− I would perform a nutritional screening as in all patients	92 (79.3)
− I would prescribe hypocaloric hyperprotein supplements to reduce weight before the intervention, but ensuring an adequate level of proteins	15 (4.3)
Don’t know/no answer	5 (4.3)

## Data Availability

Study data are available from the corresponding author upon request.

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
