# Peer review of "Knowledge and Practices of Digestive Surgeons concerning Specialized Nutritional Support in Cancer Patients: A Survey Study"

_nutrients, 2022, doi:10.3390/nu14224764_

Round 1
Reviewer 1 Report
The authors undertook to develop an extremely important issue, which is the assessment of nutritional status and nutritional treatment in cancer patients, especially cancers of the gastrointestinal tract. It was a well-prepared and designed survey study.
The limitation of the study is a failure to get a randomized sample group, however, the sample size was large enough to obtain a high scientific soundness of the research. Therefore the reviewer has no critical comments on the manuscript.
Author Response
- Thank you very much for your supportive comments regarding the scientific interest of the study.
The limitation of the study is a failure to get a randomized sample group, however, the sample size was large enough to obtain a high scientific soundness of the research. Therefore the reviewer has no critical comments on the manuscript.
- This is a cross-sectional descriptive study to assess the knowledge and practices of digestive surgeons concerning nutritional Support in cancer patients. As stated in the Methods section, participating hospitals were selected at random by conglomerate sampling (each conglomerate was an autonomous community), but a method of random selection of gastrointestinal cancer surgeons (simple or stratified random sample) was not used.
- This question is already addressed in the limitations of the study in the following sentence: “The present survey had a high response rate but the lack of acquiring a randomized sample group is a limitation to generalize data to all Spanish surgeons practicing elective gastrointestinal procedures in cancer patients.”
Reviewer 2 Report
The manuscript is well written. Novelty and conceptualisation are good. Data presentation is weak without statistical analysis, with an incomplete impression. The manuscript content is too short and unattractive, with limited page numbers. To improve the scientific impact I recommend more data with additional results and statistical significance.
Author Response
- We appreciate your positive opinion of the novelty and design conceptualization of the survey. The presentation of descriptive statistics is a limitation of the study. We have added a comment in the limitations of the study: “The presentation of descriptive statistics only is also a limitation of the study, and it would have been interesting to perform further analysis to assess differences in responses according to characteristics of participants (such as age, sex, years of practice, participation in previous training activities or research projects, surgeon’s subspecialty, working with a dietitian, etc.).”
- Regarding the length of the manuscript, the body of the text is 3268 words, excluding tables (table 1: 1052 words, table 2: 644), which is within the standard recommended length of 3000-3500 words.
Reviewer 3 Report
Thank you for this interesting piece of work demonstrating a variable knowledge of nutrition in surgical consultants.
To strengthen your conclusions I would recommend you address the selection bias that may be apparent, (with any questionnaire, those interested in the topic maybe more likely to answer). This may mean your study under reports the lack of knowledge?
Future work could allow for subgroup work - does the age / experience of the surgeon or the speciality make a difference - higher incidence of mlanutrition in pancreatic or oesophago-gastric surgery - are those surgeons more familiar with nutrition than the colorectal surgeons? How many work with a dietitian? These demographics would strengthen your conclusions
The issue over the dependence on albumin and pre-albumin warrants discussion. This is concerning, especially in a surgical cohort.
Section 3 of Table 1 needs rewording.
In your opinion, how should be classified malnutrition that occurs in the surgical patient?
Would read better as:
In your opinion, how should malnutrition that occurs in the surgical patient be classified?
Overall, an interesting piece of work
Author Response
Thank you for this interesting piece of work demonstrating a variable knowledge of nutrition in surgical consultants.
- All authors would like to thank you for your suggestions, which have contributed to improve the quality of the manuscript.
To strengthen your conclusions I would recommend you address the selection bias that may be apparent, (with any questionnaire, those interested in the topic maybe more likely to answer). This may mean your study under reports the lack of knowledge?
- In the Conclusions we have added this sentence: “The possibility of a selection bias, with participants being those more interested in the topic of the survey, limitations of knowledge may be underreported.”
Future work could allow for subgroup work - does the age / experience of the surgeon or the speciality make a difference - higher incidence of malnutrition in pancreatic or oesophago-gastric surgery - are those surgeons more familiar with nutrition than the colorectal surgeons? How many work with a dietitian? These demographics would strengthen your conclusions
- Yes, we agree that these demographics would strength the conclusion, but these data were not collected. However, we have added this suggestion in the limitations of the study: ““The presentation of descriptive statistics only is also a limitation of the study, and it would have been interesting to perform further analysis to assess differences in responses according to characteristics of participants (such as age, sex, years of practice, participation in previous training activities or research projects, surgeon’s subspecialty, working with a dietitian, etc.).”
The issue over the dependence on albumin and pre-albumin warrants discussion. This is concerning, especially in a surgical cohort.
- We have added this paragraph in the Discussion: “Finally, albumin and prealbumin are parameters longer used in surgery to identify changes in nutritional status of the patients. However, they are not accurate enough to define malnutrition, but in the present study, a decrease in serum albumin and prealbumin were the parameters that most accurately diagnosed malnutrition according to 74.1% of participants. The guidelines of the European Society for Clinical Nutrition and Metabolism (ESPEN) [28] indicate that these parameters are more related to prognosis for postoperative complications and impaired nutritional status, although a persistently low or even decreasing or increasing serum albumin concentration may be indicative of whether recovery is successful or not. In the last updated ESPEN clinical guidelines, the working group agreed that hypoalbuminemia reflects a disease-associated catabolism and disease severity rather than undernutrition [29].”
- References #28 and #29 is included.
Section 3 of Table 1 needs rewording.
In your opinion, how should be classified malnutrition that occurs in the surgical patient?
Would read better as:
In your opinion, how should malnutrition that occurs in the surgical patient be classified?
- We have modified rewording of this question as kindly suggested.
Overall, an interesting piece of work
- Thank you very much.